# Environmentally adaptive MOF-based device enables continuous self-optimizing atmospheric water harvesting

Husam A. Almassad[1], Rada I. Abaza[1], Lama Siwwan[1], Bassem Al-Maythalony[1] & Kyle E. Cordova [1] ✉

Harvesting water vapor from desert, arid environments by metal-organic framework (MOF) based devices to deliver clean liquid water is critically dependent on environment and climate conditions. However, reported devices have yet been developed to adapt in real-time to such conditions during their operation, which severely limits water production efficiency and unnecessarily increases power consumption. Herein, we report and detail a mode of water harvesting operation, termed 'adaptive water harvesting', from which a MOF-based device is proven capable of adapting the adsorption and desorption phases of its water harvesting cycle to weather fluctuations throughout a given day, week, and month such that its water production efficiency is continuously optimized. In performance evaluation experiments in a desert, arid climate (17–32% relative humidity), the adaptive water harvesting device achieves a 169% increase in water production (3.5 $L_{H2O}$ $kg_{MOF}^{-1}$ $d^{-1}$) when compared to the best-performing, reported active device (0.7–1.3 $L_{H2O}$ $kg_{MOF}^{-1}$ $d^{-1}$ at 10–32% relative humidity), a lower power consumption (1.67–5.25 kWh $L_{H2O}^{-1}$), and saves time by requiring nearly 1.5 cycles less than a counterpart active device. Furthermore, the produced water meets the national drinking standards of a potential technology-adopting country.

Six billion people will face water insecurity by 2050 as a direct consequence of climate change, improper safeguarding of water resources, expansion of irrigated agricultural practices, increased exportation of water-intensive commodities, and a growing human population[1–3]. When attempting to tackle the global water crisis, the traditional approach has been to pair national water supplies with water user needs without taking into account total water demands[1,4]. Although there is enough freshwater available to meet such demand annually on a global level, geographic and temporal variations of water demand and availability are large, meaning that water scarcity occurs and changes during specific times of the year[1,4]. An ideal first step would be for water-scarce countries to lower dependency on external water resources and develop policies to import water-intensive commodities that otherwise deplete supplies or cannot sustainably be produced domestically[1,5]. Indeed, this would be impactful; from

1996–2005, almost one-fifth of the global water footprint was dedicated for export rather than domestic consumption[5]. In parallel, during those times of water scarcity, alternative means of water reclamation, creation, production, and/or delivery must be pursued and realized, which include systemic reduction of non-revenue water, desalination, wastewater treatment, and reuse, and water harvesting in different forms[6–8]. Though these have all proven successful to varying degrees in complementing the delivery of water to meet domestic demand under different environmental and climate conditions, one emerging technology−adsorbent-based atmospheric water harvesting−stands out for its proven potential in capturing, collecting, and condensing water vapor in climate conditions where its concentration is low (i.e., desert, arid regions)[9–11]. In adsorbent-based atmospheric water harvesting, metal-organic frameworks (MOFs), a class of extended, porous crystalline materials, reign supreme given their ideal water uptake behavior

[1]Materials Discovery Research Unit, Advanced Research Centre, Royal Scientific Society, Amman 11941, Jordan. ✉e-mail: kyle.cordova@rss.jo

and capacity at relevant relative humidities (RH), favorable kinetics and thermodynamics of physisorption, and hydrolytic stability[12–16].

When employing MOFs, as well as other adsorbent materials, for atmospheric water harvesting, two modes of operation have been reported for devices that exploit their use[17]. The first is a passive mode, in which water is generated by exposing a MOF bed to atmospheric air during the night when RH is at its maximum[18–21]. During the day, when the RH is minimal, heat generated by sunlight is used to desorb the water from the MOF where it is then condensed on the surrounding walls of the passive device. The passive mode is effectively one 24 h adsorption-desorption cycle and its performance is dependent on the uptake capacity of the employed MOF at a given RH. When using MOF-801 (37 wt% water uptake capacity at 30% RH), an ideal passive device functioning at 100% efficiency would yield 588 mL$_{H2O}$ kg$_{MOF-801}^{-1}$ d$^{-1}$ at 30% RH[18–22]. Reported water generation values range from 100 to 300 mL$_{H2O}$ kg$_{MOF-801}^{-1}$ d$^{-1}$, which means that the passive device is operating at an efficiency of <51% of its capacity. To achieve water generation that is suitable for meeting a person's daily needs (≥3.5 L), considerably more material must be used (e.g., 12–35 kg$_{MOF-801}$). This has consequences with respect to device enclosure geometry and the size of the glass concentrators required for the desorption stage of the cycle. For example, to generate 3.5 L of water at the higher 68% RH, the surface area of the physical enclosure of a single- or dual-stage passive device would need to be 10.3 or 4.54 m$^2$, respectively, which is simply too large and not practical[18–21].

The second mode of operation is an active one, whereby water is generated continuously from air via multiple adsorption-desorption cycles occurring in a given day[23]. In the active mode, the adsorption phase starts with forcing air through the material bed using a fan for a set period of time. Desorption by an additional heat source occurs at any time to release the captured water and a vapor compression-refrigeration system is used to condense the water vapor. The active mode is dependent upon the dynamic water capacity of the employed MOF, which, in turn, impacts the cycling rate and quantity of water generated on a daily basis[23]. Reported active devices can generate up to 1.3 L$_{H2O}$ kg$_{MOF}^{-1}$ d$^{-1}$ at 32% RH and 27 °C—a quantity that is four-fold larger than can be achieved by a passive device under the same conditions[23]. Although this is a significant improvement over the passive mode of operation, water generation remains insufficient per kg$_{MOF}$ used for meeting daily consumption needs.

Herein, we report the design engineering of a water harvesting device using MOF-801 that extends the modes of operation beyond passive and active to 'adaptive water harvesting' (Fig. 1). This mode of operation builds upon the previous active one whereby the device carries out multiple water harvesting cycles (WHC; at times simply referred to as 'cycle') per day, but with the critical difference being that the adaptive mode optimizes the timing and efficiency of each WHC based on real-time external environmental conditions. In performance evaluation experiments in a desert, arid climate (17–32% RH), the adaptive device had a >169% increase in water production (3.5 L$_{H2O}$ kg$_{MOF-801}^{-1}$ d$^{-1}$) when compared to the best-performing, reported active device (0.7–1.3 L$_{H2O}$ kg$_{MOF}^{-1}$ d$^{-1}$ at 10–32% RH), a lower power consumption (1.67–5.25 kWh L$_{H2O}^{-1}$), and saves time by requiring nearly 1.5 cycles less per d than a counterpart active device[23]. Furthermore, we demonstrate the adaptive device's ability to continuously and consistently produce water with no loss in performance after more than 1 yr of operation. Finally, a full panel water analysis was performed to assess and then certify, that the produced water met the national drinking standards of a potential water harvesting technology-adopting country (Jordan).

## Results and discussion
### Design strategy behind the water harvesting device
To extend the scope of adsorbent-based water harvesting, we constructed a three-compartment, modular device that builds upon the

active mode of operation. As will be demonstrated, the design plays a significant role in improving the water harvesting process performance. The first compartment, termed the air intake compartment, contains an air filter that prevents solid particulate matter with size >10 nm from entering the device and a fan to push the external air through the device (Fig. 1a). An electric heater was placed next to the fan such that during the desorption phase of a given cycle, heated air can provide the necessary energy to release the adsorbed water from the MOF pores and carry that desorbed water vapor at a higher capacity to the condenser. Indeed, air has a maximum water content of 130 g m$^{-3}$ at 60 °C and 17.3 g m$^{-3}$ at 20 °C, therefore, heating the air leads to a more than sevenfold increase in the ability of the air to transport the desorbed water vapor[24]. The first RH and temperature sensor was placed in front of the electric heater to measure the air before it passes through the MOF material (see Supplementary Note 1).

The second compartment, termed the sorption compartment, was connected directly to the air intake compartment in series and contained eight trays (with room to add more) placed in parallel to the airflow (Fig. 1). Each tray was lined with aluminum (2 mm thickness) to increase heat transport and facilitate the desorption of water. Aluminum has a much higher thermal conductivity ($K_{Al}$ = 205 W m$^{-1}$ K$^{-1}$ versus $K_{acrylic}$ = 0.2 W m$^{-1}$ K$^{-1}$) and, when applying Fourier's Law, will have a 1000-fold higher heat conduction in reaching the same temperature difference as compared to acrylic[25]. For this device, MOF-801 was chosen for use given its high water uptake of 22.5 and 37 wt% at 10 and 30% RH, respectively, and its suitable α inflection point ($P/P_0$ = 0.07) that highlights the material's ability to adsorb water in arid environments[22]. Furthermore, water adsorption by MOF-801 is fully reversible with no hysteresis observed upon desorption and its mild water adsorption enthalpy of 60 kJ mol$^{-1}$ means that it does not incur a large energy penalty upon regeneration (see Supplementary Note 2)[22]. The sorption compartment was constructed to hold from 100 g to 10 kg of MOF-801 for performance measurements and has two sensors placed at the top and bottom.

The third compartment, termed the condensation compartment, was connected directly to the sorption compartment, and contained a condenser that serves as the evaporator of a larger vapor compression-refrigeration cycle. The condenser is fit within a funnel that is used to collect the condensed liquid water and transport it to the mineralization and filtration unit below (Fig. 1). A 5 mm outlet hole was incorporated at the bottom of this compartment to prevent any back pressure, friction loss, or pressure drop. In addition, this hole not only ensures a continuous flow of incoming hot air, but also forces the condensed water to the mineralization and filtration unit. The condensation compartment has one sensor that is placed before the condenser to take RH and temperature measurements of the air after it has been exposed to the MOF-801 bed (Fig. 1b).

### Performance evaluation of the active water harvesting device
After constructing the device, we then sought to test its performance in generating water at low RH via an active mode of operation. An active water harvesting cycle (WHC) was defined as follows: 40 min adsorption phase (an adsorption phase is defined as simply pulling air by the fan from the external environment through all three compartments) followed by a 20 min desorption phase (a desorption phase is defined similarly to the adsorption phase but with the electric heater turned on) that was repeated twice at which point the regeneration phase was carried out before re-starting the next WHC (regeneration is defined as 15 min of airflow without heat, 10 min with no cooling of the condenser to collect the water droplets, and 5 min with no fan to equilibrate the device before starting the next WHC). It is noted that the timings for each phase were concluded based on preliminary measurements that provided visual indications of water production as a function of time (see Supplementary Note 3). Based on this active

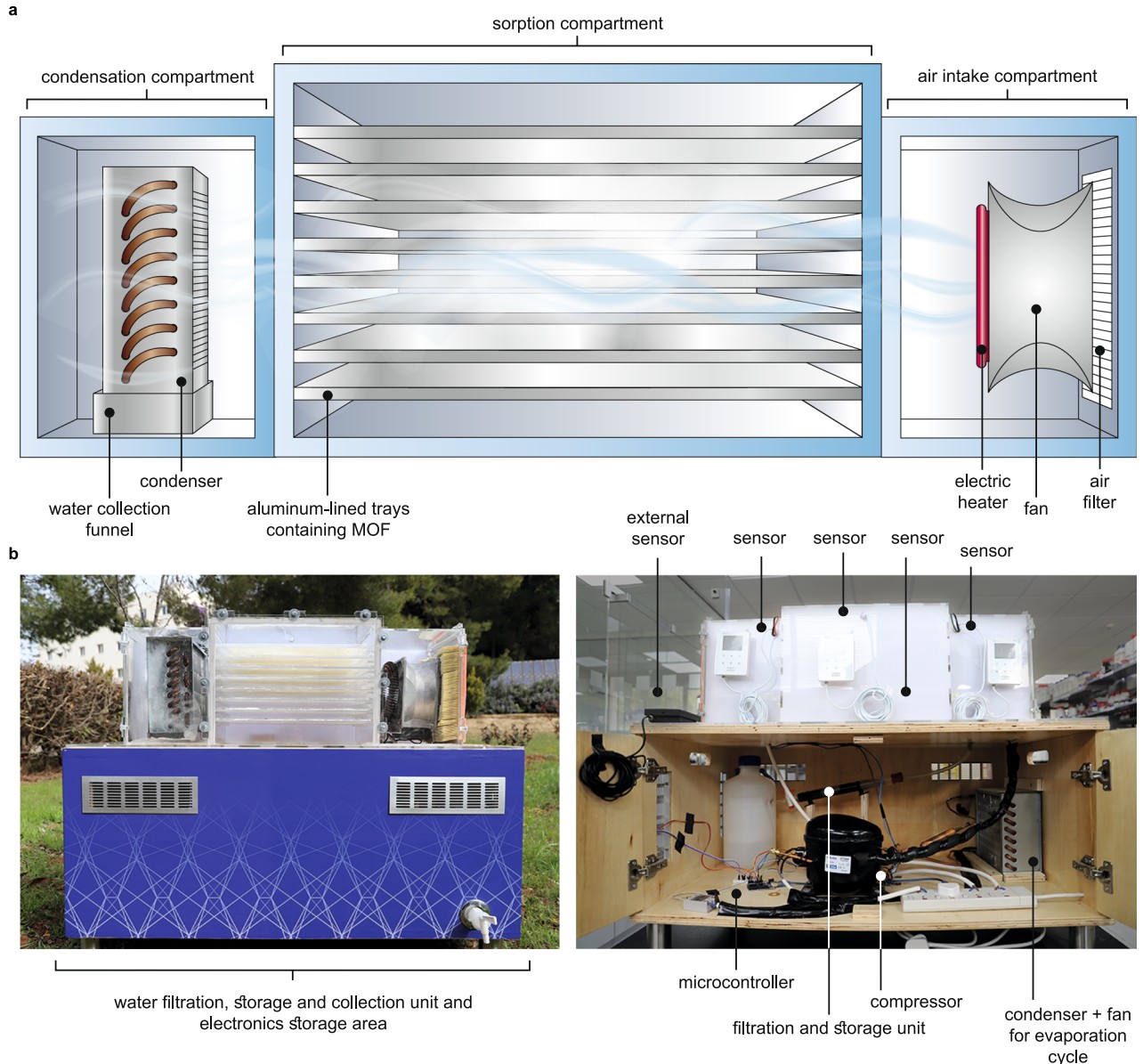

**Fig. 1 | MOF water harvester device operating in adaptive mode. a** Schematic of the water harvester device detailed herein, which is comprised of three modular compartments: air intake compartment that houses an air filter, fan, and electric heater; sorption compartment that holds the MOF-801 adsorbent on aluminum-lined trays; and condensation compartment that contains a condenser and a water collection funnel. **b** Photographs of the water harvester device with labels to different critical components.

WHC definition, our water harvesting device was examined over a range of environmental conditions (10−70% RH and 15−35 °C) and was proven capable of producing 1.2−2.6 L of water daily with an energy consumption of 3−7 kWh L$^{-1}$ (see Supplementary Note 3).

The importance of using MOFs for functional, efficient atmospheric water harvesting is their ability to effectively concentrate water vapor in air resulting in an increased dew point[13–15]. It is important to point out here that previous literature have focused exclusively on RH as the basis for water production without considering the effect of temperature[9–11,18–21]. In fact, for atmospheric water harvesting processes, both RH and temperature must be considered through the dew point value. Indeed, dew point reflects actual water quantity in the air at any given moment and, therefore, all further discussions will rely on the dew point as the basis for water production. This concept is readily visualized when plotting the desorption data from the sensor in the condensation compartment for the trial runs of our MOF-801-based device on a psychrometric chart at 101.3 kPa. As shown in Fig. 2, when

the device is operating with external conditions of 30% RH and 22 °C, the dew point corresponds to 3.6 °C. After the adsorption phase is completed, to initiate water desorption from the pores of MOF-801, heating was applied for 10 min leading to an increase in the dew point to 8.3 °C. Furthermore, an additional 10 min of heating led to an even higher dew point of 11.2 °C. This seemingly small rise in dew point has a large impact on condenser power consumption when compared with direct cooling of air without using MOF-801 (i.e., 55% reduction) (see Supplementary Note 3). Any extra heating beyond this point is superfluous and can be considered as an energy loss as such heating does not lead to a further increase in the dew point and thus signals the end of the desorption phase (20 min total heating). This begs the question: if the external conditions differ from those measured here, what impact does that have on the prescribed time for carrying out the adsorption and desorption phases of a given WHC? Indeed, it is easy to imagine that these sorption phases of the device are not adaptive nor are optimized to a wide range of conditions.

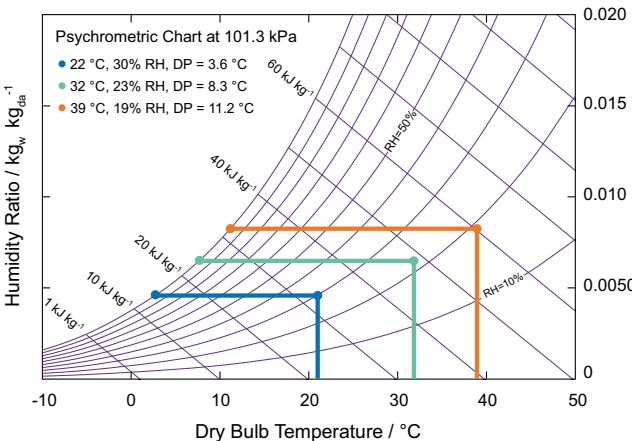

**Fig. 2 | The effect of RH and temperature on dew point when using MOF-801.**
The condensation compartment response for the desorption phase of a WHC
shown on a psychrometric chart at a starting point whose environmental condi-
tions were 30% RH and 22 °C (blue). Optimizing the heating for 10 (green) and
20 min (orange) increases the dew point by 4.7 and 7.6 °C, respectively. This
increase in dew point has a large impact on condenser power consumption. Source
data are provided as a Source Data file.

## Adaptive water harvesting

Based on the limitations of the active mode, we then sought to develop
a third mode of operation, termed 'adaptive water harvesting'. The
fundamental concept of this adaptive mode is to measure the external
environment RH and temperature and combine them together through
the dew point value. The water harvesting device was then pro-
grammed to read this measured dew point in real-time and instruct its
adsorption and desorption phase operations to react according to that
value. In principle, this will yield the highest water quantity because the
cycling rate is adaptable and maximized for productivity per time of d.

## Adsorption phase of an adaptive water harvesting cycle

The first step in developing the adaptive device was to understand the
time needed for MOF-801 (400 g) to reach full saturation at any given
environmental condition. To do this, a full desorption process was
conducted for MOF-801 by forcing hot air at 80 °C through the sorp-
tion compartment for 2 h. MOF-801 was then exposed to air with
severe conditions (35–40 °C and 15, 18, and 26% RH) and the dew point
response at the condenser compartment was recorded. As shown in
Fig. 3a, the dew point decreases to a steady-state value as a function of
time before steadily returning back to a value based on the external
environmental conditions (see Supplementary Note 4). Given that the
difference between the starting and the steady-state dew point values
reflects the quantity of adsorbed water by MOF-801, the required
timing of the adsorption phase can be elucidated. For example, at 26%
RH and a dew point temperature of 14 °C, the air has an absolute
humidity (i.e., actual water quantity in the air) of 11.4 $g_{H2O}$ m$^{-3}$. If the
adsorption time is extended to 51 min, a steady-state dew point value
of 3.4 °C is reached with an absolute humidity of 5.29 $g_{H2O}$ m$^{-3}$, which
correlates to 300 $g_{H2O}$ adsorbed within MOF-801 over that time
(Fig. 3a). Reducing the adsorption time to 21 min, which is the start of
the steady-state dew point value (at 21 min the dew point = 5.1 °C), then
by the same reasoning, 280 $g_{H2O}$ is adsorbed within MOF-801. Though
this is 7% less quantity of water adsorbed, the timing difference is
significant. Furthermore, the starts of the steady-state dew point
values for measurements at 18 and 15% RH were also identified with
respect to time (17.5 and 15 min) and correlated to the quantity of
adsorbed water (220 and 180 $g_{H2O}$, respectively). We note that closing
the 5 mm outlet in the condensation compartment forces air back to
the air intake compartment, leading to a 12–27 min increase in the

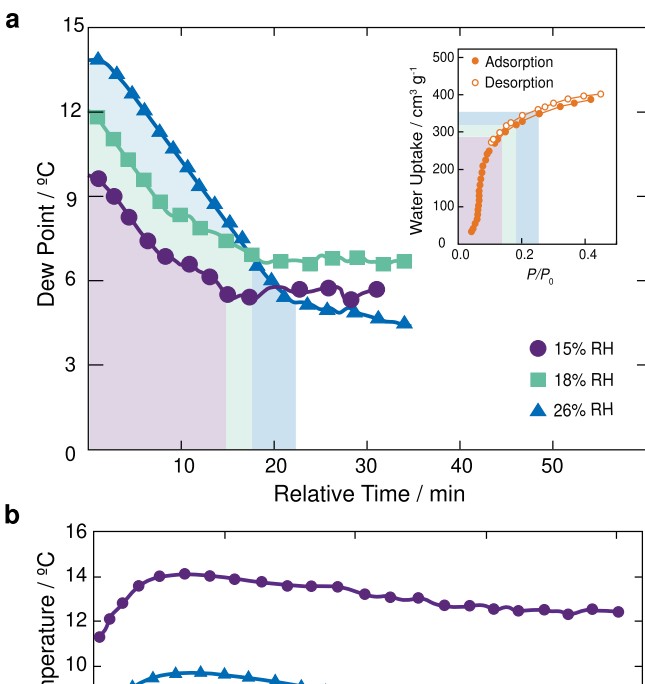

**Fig. 3 | Adsorption and desorption phases were used to develop the adaptive
WHC algorithm. a** The condensation compartment dew point response for the
adsorption phase in a WHC. Shown are the relative adsorption time needed for a
range of environmental conditions: 15 (purple circles), 18 (green squares) and 26%
RH (blue triangles). These adsorption measurements were conducted over a tem-
perature range of 35–40 °C. The inset is the water sorption isotherm for MOF-801 at
25 °C with the water uptake capacity highlighted according to those same envir-
onmental conditions. **b** The desorption phase dew point response as a function of
relative heating time recorded in the condensation compartment for various
environmental conditions: 14 (orange stars), 30 (pink squares), 34 (blue triangles),
and 45% RH (purple circles). These desorption experiments were conducted over a
temperature range of 20–25 °C. Source data are provided as a Source Data file.

adsorption time. As the profile of the water sorption isotherm for MOF-
801 (Fig. 3a inset) indicates that total uptake saturation occurs at ca.
40% RH, the measured RHs in this experiment provide satisfactory
representation of the material's adsorption performance. From this
data, an algorithm was developed to ensure that the adsorption phase
of the WHC operates using the minimal required adsorption time
needed to reach the start of the steady-state dew point at any envir-
onmental condition (see Supplementary Note 4).

## Desorption phase of an adaptive water harvesting cycle

Monitoring the timing of the desorption phase is relatively straight-
forward. The device was prepared for this measurement by carrying
out one adsorption phase to ensure that MOF-801 (400 g) was fully
saturated with water. Upon beginning the desorption phase and
monitoring the sensor placed at the condensation compartment, MOF-
801 released water vapor from its pores by observing the dew point of
the air increasing to a maximum. Reaching this maximum ultimately
signaled the end of the desorption phase, at which time a gradual

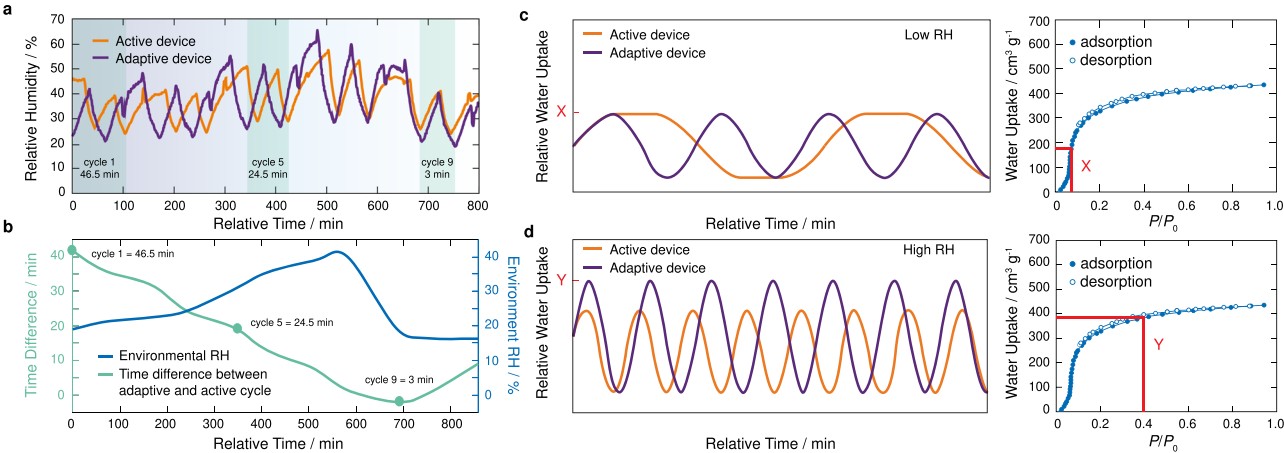

**Fig. 4 | Performance evaluation of adaptive water harvesting. a** Relative humidity (RH) response for the active (orange) and adaptive (purple) devices operating under the same environmental conditions (20–46% RH) during an arbitrary portion of the same day. The cycle time demonstrates the optimized performance of the adaptive device throughout the day with less time consumed per cycle. **b** Time difference profile between cycles carried out by the adaptive and active devices with changing environmental RH throughout the day. As environment RH increases (blue), the adaptive and active cycle time difference decreases (green). **c** The conceptual principles underlying the differences in cycling rates for the adaptive and active devices. In low RH, the pre-programmed adsorption phase of the active device (orange) is wasted after reaching the maximum uptake capacity of the material. In the adaptive device (purple), once the maximum uptake capacity of the material is realized during the adsorption phase, the desorption phase immediately starts. **d** In high RH, the timing of an active device's pre-programmed adsorption phase (orange) is less than the time required to reach the material's maximum uptake capacity. The adaptive device increases the adsorption phase time to achieve the material's maximum uptake capacity to produce more water. The water adsorption isotherm for MOF-801 at 25 °C is provided as insets to correlate potential uptake capacity under low (red X) or high (red Y) RH with cycling rates for the adaptive and active devices[22]. Source data are provided as a Source Data file.

decrease was observed until this internal dew point value was equivalent to the external dew point value (Fig. 3b). We note that the desorption phase was tested at low temperatures (20–25 °C) and varying RH (14, 30, 34, and 45% RH), from which all measurements exhibited the same behavior, but with different rates of change in dew point. In general, a higher RH resulted in higher water uptake and required a longer desorption time. According to the water sorption isotherm of MOF-801, ca. 79% of the total uptake capacity is reached by 20% RH. This means that at RH > 20%, the timing of the desorption phase is relatively the same, but significantly different at RH < 20% (Fig. 3b). Similar to the adsorption phase, when closing the 5 mm outlet in the condensation compartment, a 14–32 min increase in the desorption time was observed, which resulted in a reduction of 7–19 mL cycle⁻¹ water production. From these measurements, a second algorithm was developed to correlate heating time to the external conditions (i.e., RH and temperature) and the power (W) of the electric heater employed (see Supplementary Note 4).

**Performance evaluation of the adaptive water harvesting device**

Using RH and temperature data from sensors positioned throughout the enclosure, the device applies the two adaptive adsorption and desorption phase algorithms to control the timings of these phases as a function of external environmental conditions (i.e., RH and temperature). To evaluate the performance of this adaptive mode of operation against the previous active mode, we loaded two identical devices with 400 g of MOF-801 and programmed the operation of each device according to Supplementary Figs. 26 and 31. Under the same conditions, the performance for both devices in terms of change in RH as a function of time is presented in Fig. 4a, b for a 14 h total time spanning a portion of the day into the night. Over this period, the lowest and highest external RHs were measured to be 19% and 46%, respectively. Interestingly, the first cycle of the adaptive device with external conditions of 23% RH and 25 °C was found to be 46.5 min faster than the active device demonstrating its ability to adapt and enhance the WHC rate.

When considering an adsorbent material's water uptake capacity is limited by RH regardless of extended cycling time, the advantages of the adaptive mode of operation become clear. Under low RH conditions, the adaptive device will reach its maximum capacity at the specified RH, at which point it will immediately begin the desorption phase without consuming more time in the adsorption phase. However, for the active device in these same conditions, the adsorbent will reach its maximum capacity faster. Therefore, the remaining pre-programmed time for the adsorption phase is wasted. Similar reasoning can be extended to the desorption phase. Lower water uptake per cycle in the adaptive mode of operation requires a shorter desorption time than the pre-programmed desorption time used in the active mode of operation (Fig. 4c).

As the RH increases throughout the day and into the night, the difference in cycling rates becomes indistinguishable (e.g., <3 min for cycle 9 at 46% RH and 25 °C). This is because an increased RH during the night increases MOF-801's water uptake, which, in turn, increases the timings of the adsorption and desorption phases of the adaptive device (Fig. 4a). Under high RH conditions, the adaptive mode of operation will reach MOF-801's maximum capacity and, once achieved, will immediately start the desorption phase. For the same conditions, the pre-programmed timing for the adsorption phase of the active mode of operation will not be enough to reach MOF-801's maximum capacity leading to lower water production (Fig. 4d).

From this comparison, the adaptive device was proven to produce 26% more water (3.52 L_H2O kg_MOF-801⁻¹ d⁻¹) than our active device (2.6 L_H2O kg_MOF-801⁻¹ d⁻¹) under the same climate conditions. For example, in cycle 3, the RH difference during the adsorption phase of the adaptive device was 39% compared with 28% calculated for the active device. This is a 36% increase in water adsorption by MOF-801 during this WHC. Similarly, in cycle 3, the RH difference during the desorption phase of the adaptive and active devices were 37% and 26%, respectively, which represents a 45% increase in desorbed water by the adaptive device. Resulting from the increased WHC rates, the adaptive device realized a 44% reduction in the consumed power per WHC under low RH conditions (20%) in comparison to the active device (see Supplementary Note 5). When operating at high RH conditions (46%), a 26% reduction in consumed power per WHC was achieved by the adaptive device with respect to the active device. Overall, the adaptive

**Table 1 | Performance comparison of reported adsorbent-based water harvesters with different modes of operation**

| Mode of operation | Adsorbent material | Proven environmental conditions (RH%) | Device size (m² or kg$_{ads}$) for 3.5 L[a] | Device size (m² or kg$_{ads}$) for 3.5 L[b] | Water/ kg$_{ads}$ (L)[c] | Water/ kg$_{ads}$ (L)[d] | Cycles (cycle d$^{-1}$) | Power (kWh L$^{-1}$) | Filtration/ mineralization | Water quality[e] | Ref. |
|---|---|---|---|---|---|---|---|---|---|---|---|
| Adaptive | MOF-801 | 10–70 | 0.32 m² or ca 1.9 kg$_{MOF}$ | 0.32 m² or ca 1.05 kg$_{MOF}$ | 1.8 | 3.5 | 6 – >30 | 1.67 – 5.25[f] | Yes | Pass | This work |
| Active | MOF-801 | 10–70 | 0.32 m² or ca 2.9 kg$_{MOF}$ | 0.32 m² or ca 1.45 kg$_{MOF}$ | 1.2 | 2.4 | 18 | 3 – 7[f] | Yes | Pass | This work |
| Active | MOF-303 | 10–32 | ≥5 kg$_{MOF}$ | ≥2.9 kg$_{MOF}$ | 0.7 | 1.3 | 9 | -- | No | -- | 23 |
| Passive | MOF-801/ AQSOA-Z01 (zeolite) | 10–40 | 73.5[g]/32.4[h] m² | 10.3[g]/4.54[g] m² or 12[g]/35[h] kg$_{MOF}$ | 0.1 | 0.3 | 1 | -- | No | -- | 18–21 |
| Passive | Alg-CaCl$_2$ | 25–70 | ≥4.38 kg | -- | 0.8 | -- | 1 | -- | No | -- | 26 |
| Passive | SMAG[i] | 60–90 | -- | 4.17 m² | -- | 0.84 | 1 | 1000 W m$^{-2}$ [j] | No | -- | 27 |
| Passive | SHPF[k] | 10.6–41.6 | ≥8.1 kg | ≥3.89 kg | 0.43 | 0.9 | 1 | 4.1 | No | -- | 28 |

[a]Environmental conditions for producing 3.5 L of liquid water = 10–30% RH.
[b]Environmental conditions for producing 3.5 L of liquid water = 30–60% RH.
[c]10–30% RH.
[d]30–60% RH.
[e]Meets a country's national drinking standards.
[f]Dependent on environmental conditions.
[g]Single-stage device.
[h]Dual-stage device.
[i]SMAG = super moisture-absorbent gels, super hygroscopic polymer films.
[j]Solar flux for desorption.
[k]SHPF = super hygroscopic polymer films.

device consumed 1.67–5.25 kWh L$_{H2O}^{-1}$ based on the weather conditions of that specific day (19–46% RH) and saved considerable time by requiring 1.5 cycles less than the active device. Figure 4b shows the variation of the cycling time as the RH varies during the day.

Compared with other adsorbent-driven water harvesting devices, our MOF-801-based adaptive device reduces the amount of adsorbent material required for producing enough water required to fulfill daily personal consumption needs (3.5 L) by 75%, 73%, and 57% with respect to devices based on super moisture-absorbent gels, super hygroscopic polymer films, and hygroscopic salts in a hydrogel-derived matrix, respectively (Table 1)[26–28]. Furthermore, our adaptive device drastically reduces energy consumption by 60% compared to the benchmark device based on super hygroscopic polymer films while maintaining a smaller physical footprint (Table 1).

**Dependency of MOF quantity on water production**
It is well-known that water production is directly dependent on MOF quantity in water harvesters operating in the passive mode. However, direct dependence for the active mode of operation is unclear and is yet to be proven because of the limitations of the compartment size and the power consumption requirements from increasing that size. To assess this dependency in the adaptive mode of operation, a control experiment was performed whereby the water production output was correlated to the MOF-801 quantity loaded into the device. Specifically, the device was gradually loaded with MOF-801 starting from 0 g (empty device) to 100, 200, and 400 g and subsequently operated under controlled environmental conditions of 20, 30, and 40% RH. Water production for the device loaded (or not) with each amount was measured three times for 24 h for each controlled climate condition with the average output presented in Supplementary Fig. 32. At 50% RH and 400 g$_{MOF-801}$, the device produced 870 mL$_{H2O}$ compared with 420 and 190 mL$_{H2O}$ when using 200 and 100 g$_{MOF-801}$, respectively. When the device was empty (0 g of MOF-801), the condenser was set to 8–11 °C (i.e., the dew point when using MOF; Fig. 2), which led to a water production of <2 mL. A similar trend was observed for the other measured conditions and the dependence of MOF quantity on water production was established.

Based on this experiment, our device requires ca 1.6 kg$_{MOF}$ to produce enough water to meet personal consumption needs, which is significantly less than what is needed for reported MOF-based active

water harvesting devices (≥2.9 kg$_{MOF}$ required) and the 12–35 kg$_{MOF}$ needed for passive devices to achieve the same production amount (Table 1). When taken together with compartment size considerations, our device's physical design is clearly advantageous.

**Long-term performance**
Long-term water stability and sorption capability are pre-conditions for using MOFs in water harvesting applications[29]. Although the physicochemical properties of MOFs play a critical role in determining their lifetime, the device engineering must be considered as a potential influencer (positive or negative) of the MOF's properties and resulting lifetime for use. For example, the device can optimize heat transfer and airflow, which will lessen the strain on the MOF material. Furthermore, the adaptive mode of operation ensures that cycling rates result in maximum performance output while not overextending material stability. To demonstrate our adaptive device's long-term performance, we performed a 24 h stress test under extreme conditions (22% RH and 25 °C) after the device had performed >1000 cycles (equivalent to ca. 1 yr of operation). As shown in Fig. 5a, the adaptive device produced an appreciable amount of water per WHC (40 mL cycle$^{-1}$). To confirm the structural stability of MOF-801, powder X-ray diffraction analysis was performed after this stress test was completed, which confirmed that the crystallinity of MOF-801 was retained with the diffraction pattern matching that of the simulated one from the single crystal structure (see Supplementary Note 6).

**Practical production of water from desert climates**
In general, atmospheric water harvesting in desert, arid regions is difficult, if not practically impossible for other conventional water harvesting technology, due to the low relative humidity[17,30]. This is because conventional water harvesting techniques (e.g., direct cooling of air, condensation, fogging) require an enormous amount of energy to be effective under these climate conditions[8,31]. The attraction of MOF-based water harvesting is the fact that these materials can selectively capture water vapor at low RH, concentrate and release it to increase the dew point for condensation purposes[17]. Therefore, in order to demonstrate the utility of our adaptive device, we conducted an experiment whereby the device (400 g of MOF-801) was exposed to natural, desert air in Amman, Jordan (17% RH and 25 °C) for 24 h.

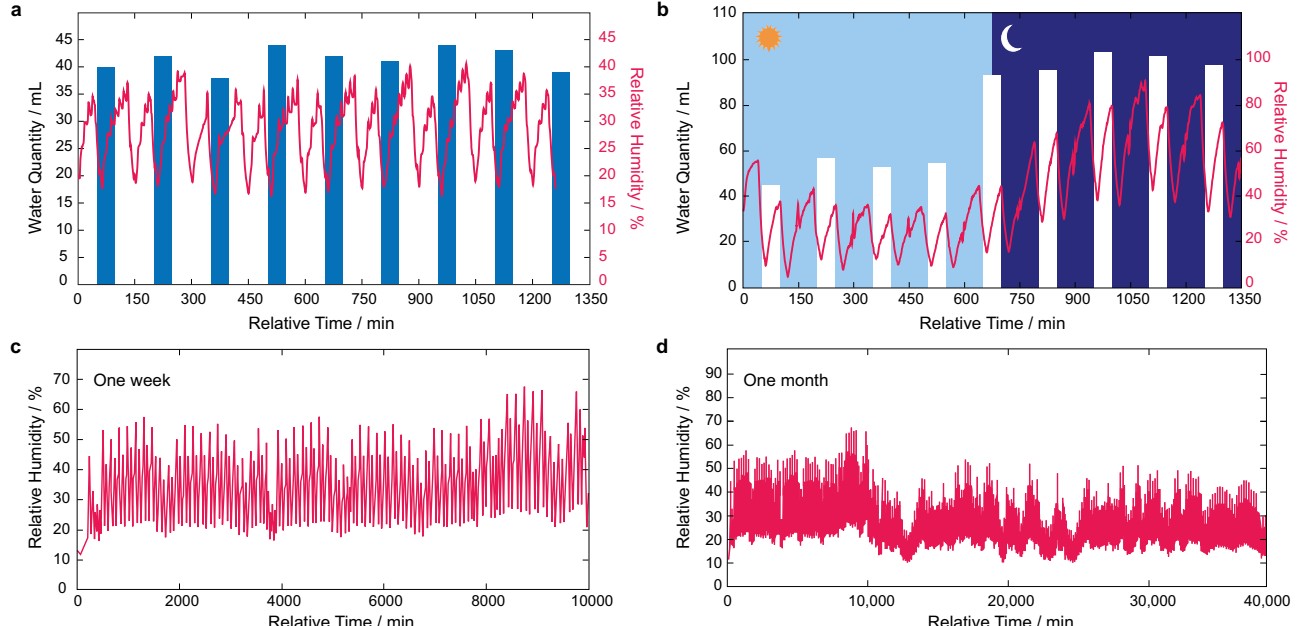

**Fig. 5 | Adaptive water harvesting stress tests for a day, week, and month in real desert, arid environments. a** Stress test for 24 h performed under a controlled environment (20% RH and 25 °C). The adaptive device is consistent in water production without loss in performance. **b** Real-environment performance evaluation for one full day demonstrating adaptive performance as a function of day-night fluctuations in environmental conditions. **c** Real-environment continuous performance for one week. **d** Monitoring the condensation compartment response to demonstrate the adaptive device's performance for one month of continuous operation. All data was collected on the adaptive device after it had performed >1000 cycles, which is equivalent to ca. 1 year of operation. Source data are provided as a Source Data file.

From this experiment, we observed that during the low RH of the desert day, the adaptive device produced 52 mL cycle⁻¹, which is significant given the extreme desert climate conditions. The production amount is doubled per WHC during the higher RH conditions (50–60% RH) of the desert night (105 mL cycle⁻¹). To push this experiment further, we continuously operated the device in a desert, arid environment for 1 continuous week that was extended to >1 continuous month. As depicted in Fig. 5b–d, the adaptive device effectively responded to the daily changes in weather conditions and continuously produced appreciable amounts of liquid water under severe desert conditions.

## Full water quality analysis

Before the condensed water can be used for personal consumption, the water was gravity-filtered and mineralized through a column containing alternating layers (2–3 mm) of activated carbon, sand, and limestone. A full water analysis panel was then carried out for the collected water to assess whether it met the national drinking standards of Jordan. The first analysis followed the Standard Methods for the Examination of Water and Wastewater (SM 3120-B, 4110-B, 3111-B, and 31112-B), in which metal identification and concentration were determined by inductively coupled optical emission spectroscopy. Specifically, the water was analyzed for: Al (<0.1 ppm), Na (5.7 ppm), Pb (<0.01 ppm), Cd (<0.003 ppm), Cu (<0.05 ppm), Mn (<0.05 ppm), As (<0.01 ppm), Zn (3.58 ppm), Fe (<0.1 ppm), Sb (<0.002 ppm), Mo (<0.01 ppm), Zr (<0.1 ppm), B (<0.1 ppm), Se (<0.04 ppm), Ba (<0.1 ppm), Cr (<0.02 ppm), Hg (<0.001 ppm), Ag (<0.1 ppm), and Ni (<0.05 ppm). As indicated, all were less than the specification limits of the national drinking standards. The volatile organic compound content was then determined following the same standard methods. For this, the concentrations of benzene (<10 ppm), total xylene (<20 ppm), trichloroethene (<20 ppm), tetrachloroethene (<20 ppm), ethylbenzene (<20 ppm), and toluene (<20 ppm) were determined to be less than the specification limits. Finally, microbiological analysis (SM 9213 E, 9230 B, 10200, 9215 AB) was carried out, in which the content of pseudomonas aeruginosa (<1 colony-forming unit 100 mL⁻¹; CFU), fecal streptococci

(<1.1 most probable number; 100 mL⁻¹; MPN), fecal enterococcus (<1.1 MPN 100 mL⁻¹), algae type and count (not seen); and heterotrophic plate count ($8.3 \times 10^2$ CFU mL⁻¹) were determined to meet the national standards. Detailed results and certification of the collected water for meeting Jordanian standards of drinking water are provided in Supplementary Note 7.

## Cost analysis

Based on a theoretical production of 100 devices, we calculated the cost to produce a single device that is loaded with 1 kg of MOF-801 to be ca. $625 USD (see Supplementary Note 8). Assuming a 10-year device lifespan and taking the average water production for the adaptive water harvesting device per day (2.65 L), at this production cost, the off-grid price per $L_{H2O}$ produced is projected to be 6.4 US cents and the device cost per day of use is 17 US cents. Given that the adaptive water harvesting device consumes 1.67–5.25 kWh L⁻¹ to produce 1.8–3.5 $L_{H2O}$ d⁻¹ and the kWh price in Jordan is ca. $0.10 USD, the on-grid price per $L_{H2O}$ ranges from $0.17–0.53 USD, which, although seemingly high, remains lower than commercial water sources (see Supplementary Note 8).

Provided the threshold relative humidity (>10% RH), our adaptive device can operate throughout the world, with the notable exception of the Arctic Circle and the Antarctic, to serve the water needs of >2 billion people living without access to safely managed drinking water (SMDW)[32]. Therefore, to put our cost analysis into a global prospective, a random selection of countries representing lower-middle (Morocco and Nigeria) and upper-middle (Mexico and Jordan) income classification was chosen. Indeed, 31–40, 71–80, 51–60, and 11–20% of the population of Morocco, Nigeria, Mexico, and Jordan, respectively, lives without SMDW and could benefit from access to this technology (Table 2)[32]. Considering the cost of production together with on-grid electricity costs for these countries, our adaptive water harvesting device can provide financial savings of up to 49%, 63%, 63%, and 46% in water costs in Morocco, Nigeria, Mexico, and Jordan, respectively (Table 2).

**Table 2 | Price comparison of water produced by adaptive water harvesting (AWH) and commercial sources for representative lower-middle and upper-middle income countries**

| Country | Population without SMDW (%)[32] | Commercial drinking water price ($, USD)[33] | Household electrical energy cost ($, USD kWh⁻¹)[34] | Adaptive AWH drinking water cost ($, USD L⁻¹)[a,b] | Adaptive AWH drinking water cost ($, USD L⁻¹)[a,c] | Price reduction (%)[a,c] |
|---|---|---|---|---|---|---|
| Morocco | 31–40 | 0.50 | 0.12 | 0.20–0.61 | 0.26–0.68 | 49 |
| Nigeria | 71–80 | 0.42 | 0.057 | 0.095–0.30 | 0.16–0.36 | 63 |
| Mexico | 51–60 | 0.56 | 0.087 | 0.145–0.46 | 0.21–0.52 | 63 |
| Jordan | 11–20 | 0.42 | 0.10 | 0.17–0.53 | 0.23–0.59 | 46 |

SMDW = safely managed drinking water.
[a]Dependent on environmental conditions.
[b]Excluding capital asset costs.
[c]Including capital asset costs (assuming 10-year lifespan).

Variations in atmospheric conditions significantly affect the water harvesting process in terms of water production efficiency and power consumption. By developing the adaptive mode of water harvesting operation, we demonstrate how a MOF-based device could be environmentally responsive and adaptive. This led to considerable advancements in performance, most notably, a 169% increase in water productivity and a 44% reduction in power consumption compared to state-of-the-art reported devices. Moving forward, maximizing heat and mass transfer via optimized airflow dynamics (e.g., fluidized bed, shaped bodies) will lead to further exploitation of the sorption properties of MOFs for practical water harvesting. Furthermore, monitoring the performance of adaptive water harvesting devices over periods of time longer than 1 yr will provide insight into such devices' ability to respond to multiple seasonal climate changes. All in all, coupling the results from these future investigations to an adaptive mode of operation will yield a water harvesting device that can bring water security and independence to anyone, anywhere at any time.

## Methods

### Water harvesting device construction

The water harvesting device is comprised of three rectangular prismatic compartments held together by cell cast acrylic sheets (Moden Glas) as walls. One acrylic sheet is used as the base of the device ($1 \times 0.6$ m, 5 mm thickness), which extends across all three compartments. The base has regularly cut slots ($40 \times 5$ mm) to fit the walls of each compartment in place. The first compartment ('air intake compartment') is constructed from 4 cell cast acrylic sheets as walls ($280 \times 280 \times 200$ mm, 5 mm thickness). The internal front wall has rubber lining and is connected to the top and side walls by pre-designed facets that are screw-sealed to allow easy access to the compartment. This wall has a $230 \times 250$ mm window cut by a $CO_2$ laser cutter (Trotec Speedy 400), which is fitted with an air-panel filter (M Filter No. K418). The remaining top and side walls are sealed together via chloroform. Inside the air intake compartment is an electric aluminum-finned copper coil heater (500 W) and a one-phase alternating current fan (Orix No. MRS18-DC-F6) placed consecutively next to the air-panel filter. The second compartment ('sorption compartment') is directly connected to the air intake compartment and is constructed from 5 cell cast acrylic sheets as follows: (i) shared wall with air intake compartment: $290 \times 370$ mm with 8, $270 \times 8.5$ mm slots laser cut to direct airflow; (ii) back wall: $290 \times 478$ mm; (iii) top wall: $300 \times 400$ mm; (iv) front wall: $290 \times 478$ mm; and (v) shared wall with the third compartment ('condensation compartment'): $290 \times 370$ mm with 8, $270 \times 8.5$ mm slots laser cut to direct airflow. The front wall has rubber lining and is connected to the top and side walls by pre-designed facets that are screw-sealed to allow easy access to the compartment. The remaining top and side walls are sealed together via chloroform. The sorption compartment is comprised of 8 cell cast acrylic trays ($470 \times 360$ mm) each having an aluminum lining (2 mm thickness) to facilitate heat transfer. The trays are held in parallel to the

airflow by acrylic shelving that are attached beneath rows of airflow slots fastened to the two shared walls. The condensation compartment is constructed from 3 cell cast acrylic sheets ($280 \times 280 \times 200$ mm, 5 mm thickness) and has a 3-lined aluminum-finned condenser ($250 \times 250$ mm) housed inside. This condenser is fit within a stainless-steel funnel ($250 \times 100$ mm) that is used to collect condensed liquid water and transport it to the mineralization and filtration unit below. This condenser is the evaporator of a larger compression-refrigeration cycle and is, therefore, connected via copper tubing (1/4 in) to a refrigeration compressor (1/3 hp, Model No. GFF86AA; Siberia Co.), Two-lined aluminum-finned condenser with attached cooling fan ($250 \times 250$ mm; 5 W), and an expansion tube. This compressor was operated using R-134a as the working fluid (Schild Refrigerant). The condensation compartment was semi-open to the atmosphere via a 5 mm hole outlet drilled into the bottom. To monitor temperature and relative humidity throughout the device, 4 holes (20 mm diameter) were laser cut into the following walls: (i) top wall of the air intake compartment in a position of 70 mm after the heater; (ii) front wall of the sorption compartment in centered positions on the top and bottom of this wall; (iii) top wall of the condensation compartment in a position of 70 mm before the condenser. These holes were prepared to fix the sensor probes (Logitech, RCW800 Wi-Fi). One additional sensor probe was used to monitor external environmental conditions. The collected water from the condensation unit is immediately fed into a glass column ($30 \times 150$ mm; $l \times d$) that is packed with alternating layers of limestone, sand, and activated carbon. The water is subsequently gravity-filtered and mineralized and collected in a 5 L antimicrobial carboy. Detailed schematics are provided in Supplementary Note 1.

### The synthesis and characterization of MOF-801

MOF synthesis and characterization details are presented in Supplementary Note 1.

### Active water harvesting device operation

The active water harvesting mode of operation is based on a simple adsorption-desorption cycle. In a typical run, the fan and the condenser, drawing 54.5 and 184 W of power, respectively, are activated for 40 min. This provides exposure of the MOF material to external airflow to facilitate the adsorption of water vapor. The condenser was set at an arbitrary temperature well below the ambient environment's dew point. After 40 min, the heater is activated for 20 min to initiate desorption. Since the fan and the condenser remain active, this stage draws 239 W of power. The heated air reached 45 °C on average when reaching the sorption compartment containing the MOF material. Water droplets form instantly upon initiating the desorption stage of the active cycle. These are gravity collected and filtered throughout the entirety of the desorption stage and into the adsorption stage of the next active cycle. After 60 min total (40 and 20 min for adsorption and desorption, respectively), one active cycle is complete.

Theoretically, 24 active cycles can be carried out in 1 d regardless of external ambient conditions. Detailed measurement procedures are provided in Supplementary Note 3.

**Adaptive water harvesting device operation**

The environment-adaptive water harvesting mode of operation builds upon the active mode in terms of each cycle being an adsorption and desorption stage. However, the difference is in the time and responsiveness of each stage. Using historically collected data from the active device, an algorithm was developed and applied to the environment-adaptive water harvesting device by employing a uno R−3 microcontroller (Arduino), a mini breadboard (400 tie points), 3-relay modules (5 V direct-current 1-channel; Songle), and an external digital temperature and relative humidity sensor (Aosong, No. AM2315). In the environment-adaptive water harvesting mode, the adsorption stage begins with the fan and condenser, drawing 54.5 and 184 W of power, respectively, being activated for a prescribed period based on the external environmental conditions. The set condenser temperature varied as a function of the dew point calculated from the environmental conditions. The desorption stage then began by activating the heater for a prescribed period based on the adaptive algorithm. The heated air reached different temperatures according to the external environmental conditions. Water droplets form instantly upon initiating the desorption stage and were gravity collected and filtered through the entirety of the desorption stage and into the adsorption stage of the next adaptive cycle. In general, across the relative humidity scale, notably at RH < 20%, the environment-adaptive water harvesting mode can perform considerably more cycles (i.e., generate more water) than its active counterpart. Detailed measurements and data collected are provided in Supplementary Note 4.

## Data availability

The authors declare that all data supporting the findings of this study are available within the article and its Supplementary Information or from the corresponding author upon reasonable request. Source data are provided with this paper.

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

## Acknowledgements

The authors are grateful to the Royal Scientific Society for financial support of this work (K.E.C.). Further support was provided by MISTI Global Seed Funds and the MIT-Jordan Abdul Hameed Shoman Foundation Seed Fund (No. 0000000093; K.E.C.), and the Alliance of International Science Organizations (ANSO-CR-PP-2020-06; K.E.C. and B.M.). The Synchrotron-light for Experimental Science and Applications in the Middle East (SESAME; MS Beamline Nos. 20190028 and 20210003; K.E.C.) for beamtime and Dr. Mahmoud Abdellatief (SESAME) for his support. Eng. Iyad Al-Dasouqi (RSS), Eng. Bara'a Ahmed (RSS), Eng. Omar Abu Zaid (RSS), and Eng. Osama Abu Al-Hija (RSS) are gratefully acknowledged for support in securing supplies and instrument time as well as for helpful discussions.

## Author contributions

Conceptualization: K.E.C.; data curation and formal analysis: H.A.A., R.I.A., and L.S.; experimental design, validation, and investigation: H.A.A., R.I.A., L.S., and B.A.M.; project administration, funding acquisition, and resources: K.E.C. and B.A.M.; original draft: K.E.C. and H.A.A.; review and editing: H.A.A., R.I.A., L.S., and B.A.M. All authors have read and approved the manuscript and agreed to publish.

## Competing interests

A patent has been filed: Royal Scientific Society (patent applicant), K.E.C. and H.A.A. (inventors), PCT application serial no. PCT/JO2022/050012 with priority date 26 July 2021, covering several water harvesting device embodiments and active and adaptive methods of atmospheric water generation. K.E.C. is a founder and H.A.A. is an employee of Green Oasis for Research and Development, L.L.C. and AquaPoro Ventures, Ltd., which are companies pursuing commercialization of the technology reported herein. K.E.C. and H.A.A. declare no other competing interest. The remaining authors declare no competing interests.
