## [Peer Review File · Nature Communications]

Environmentally adaptive MOF-based device enables continuous self-optimizing atmospheric water harvestingREVIEWER COMMENTS

Reviewer #1 (Remarks to the Author):

This is an interesting study, I would argue primarily an engineering-based study, on the issue of water harvesting from the atmosphere. There is naturally a lot of interest in this topic these days, and I assume it will become even more important in the next few years. The work follows that from other prominent groups in using MOFs, in particular MOF_801 as the active adsorbent, but improves upon the design to produce more water than in previous studies. One of the more important contributions of this paper, in my view, is a very clear explanation and extensive cycling studies that they discuss, which go well beyond what is typical for other MOF-based materials and papers in the field of atmospheric water harvesting. In the same sense, and primarily because the merits of the paper are almost entirely metric (that is, comparisons with previous results in terms of efficiency, etc), I would like to see this device and its performance compared with the performance of the water harvesting work with polymeric gels coming out of UT Austin. An apples to apples comparison would help put this work in broader context, such that it would appeal to the broader readership rather than just the (admittedly broad) MOF community. I do recommend that some comparison be made with that system, which is also a materials-based approach, and thus relevant. Once that is done, I do think there is merit for this paper to appear in Nature Communications.

Reviewer #2 (Remarks to the Author):

This manuscript reported an environment-adaptive atmospheric water harvesting (AWH) strategy by tuning the working time of the adsorption and desorption processes according to the external RH to improve the water yield per day. The whole system is composed of an air intake compartment, a sorption compartment, and a condensation compartment. Benefiting from the optimized cycling time, the designed system can efficiently generate clean water from arid environments. While the proposed system design is somehow interesting in terms of operation optimization based on an environmental adaptive strategy, the novelty and significance of this work are rather low, since neither new scientific insights of the AWH process nor an in-depth understanding of the underlying mechanism (such as thermodynamics) was proposed. In addition, the lack of detailed explanation or discussion on some critical questions and the insufficient performance comparison with literature lead to unconvincing claims of the record high water production. Overall, this work presented an engineering optimization rather than a scientific innovation. Therefore, this current version of this work is not adequate for publishing in a prestigious journal like Nature Communications, maybe more suitable for a specific engineering journal.

Detailed comments are listed below.

1. The authors stated that "... an adsorption phase is defined as simply pulling air by the fan from the external environment through all three compartments ... a desorption phase is defined similarly to the adsorption phase but with the electric heater turned on ...", which means even during the desorption, continuous external air will be injected into the whole system. Since the continuously injected external air also reaches the condensation part and being condensed, how this part of inlet air contributes to the water yield? A control experiment identical to the desorption experiment but without MOF needs to be conducted to reveal the actual contribution of the MOF-801 to the water yield.
2. While most works in literature use a closed system during desorption (easy to reach a high RH for collection), the current work chooses to maintain an open system during desorption. Please compare the difference between the two modes and provide corresponding data if necessary.
3. In Fig 3, the dew point at the condenser compartment was used to monitor the sorption and desorption processes along with time. At the equilibrium state (no matter during the adsorption or desorption phase), no vapor will be captured or released by the MOF-801 (no net vapor flux), which means the air reached in the condenser compartment should be the same as the external air. However, unlike the desorption phase where the inside dew point is finally equivalent to the external dew point, the inside dew point at equilibrium during the adsorption phase shows a clear difference with the inlet air. Please explain why.
4. Please explain why the authors choose different conditions between adsorption and desorption

in Fig 3? The different conditions make it hard to compare the dew point change during these two connected processes.

5. In Table S1, three kinds of procedures were studied in the active device experiments and the results show more adsorption/desorption cycles in a water harvesting cycle (WHC) leads to a higher water yield per day. The authors chose the two adsorption/desorption cycles mode (i.e., Procedure 3) as the final procedure to do the following experiments. Why didn't the authors further increase the adsorption/desorption cycle numbers? Will the water yield per day keep rising along with the increased adsorption/desorption cycles? Will it lead to a higher water yield even than the adaptive strategy?

6. In Fig 4, why the curve of active operation didn't start from 0 min? The current figure is hard to read, and a summary figure compares the time per cycle and cycling numbers per day between the adaptive and active operations will help the reader to grasp their difference. In addition, an energy efficiency analysis may also be helpful in showing the advantages of the adaptive strategy over the normal active strategy.

7. The figure caption of Fig 4 is inconsistent with the figure. Please correct it and also check the whole manuscript to avoid any potential errors.

8. The author claimed "a 169% increase in water productivity and a 44% reduction in power consumption compared to state-of-the-art reported devices" in the conclusion section. Therefore, a comprehensive and systematic comparison with literature is necessary to support this claim.

Reviewer #3 (Remarks to the Author):

This contribution deals with the impressive design engineering of a new so-called 'adaptive water harvesting' device by adjusting the adsorption and desorption timing through a developed algorithm according to the atmospheric varied relative humidity during the day to optimize the harvesting efficiency and achieve a 169% increase in water production compared with the fixed-time water harvester. This is a big step to push forward the water harvester in a smart and automatic direction. Considering the novelty and significance, the present work deserves an urgent publication in Nature Comm. Several minor issues should be addressed before official publication:

- 1) The powder size and packing efficiency of MOF-801 should be presented since they are related to the timing of adsorption and desorption;
- 2) Is the power of the fan influencing the adsorption and desorption efficiency?
- 3) I am confused about the condensation compartment dew point decrease during the adsorption process if the fan is open and the system is connected to the outer space, and the system is still open during the desorption process?
- 4) How does the environment temperature influence the harvesting process?
- 5) The caption of figure 4 is not consistent with the figure.

Responses to Reviewers' Comments for Manuscript: NCOMMS-22-15900

General Points of Interest

We are delighted that two reviewers agreed about the novelty and significance of our work. Specifically, the following comments stand out:

Reviewer #1

"This is an interesting study, I would argue primarily an engineering-based study, on the issue of water harvesting from the atmosphere. There is naturally a lot of interest in this topic these days, and I assume it will become even more important in the next few years. The work follows that from other prominent groups in using MOFs, in particular MOF_801 as the active adsorbent, but improves upon the design to produce more water than in previous studies. One of the more important contributions of this paper, in my view, is a very clear explanation and extensive cycling studies that they discuss, which go well beyond what is typical for other MOF-based materials and papers in the field of atmospheric water harvesting. I do think there is merit for this paper to appear in Nature Communications."

Reviewer #2

"This is a big step to push forward the water harvester in a smart and automatic direction. Considering the novelty and significance, the present work deserves an urgent publication in Nature Comm."

Point-by-Point Responses to Comments by the Reviewers

The reviewers' comments are reproduced below in italics and our responses follow each of the comments.

Reviewer #1

1. *In the same sense, and primarily because the merits of the paper are almost entirely metric (that is, comparisons with previous results in terms of efficiency, etc), I would like to see this device and its performance compared with the performance of the water harvesting work with polymeric gels coming out of UT Austin. An apples to apples comparison would help put this work in broader context, such that it would appeal to the broader readership rather than just the (admittedly broad) MOF community. I do recommend that some comparison be made with that system, which is also a materials-based approach, and thus relevant.*

Answer: We agree with the reviewer and have carried out an apples-to-apples comparison of the MOF-based active and adaptive water harvesters reported in our work with other adsorption-driven water harvesting technologies based on super hygroscopic polymers films, super moisture-absorbent gels, zeolites, and salt composites.

Accordingly, we have revised Table 1 in the manuscript (Table 1, Page 8) and have included more discussion on the results of this comparison (Pages 7, Left Column, Paragraph 3):

“From this comparison, the adaptive device was proven to produce 26% more water ($3.52 \text{ L}_{\text{H}_2\text{O}} \text{ kg}_{\text{MOF-801}}^{-1} \text{ d}^{-1}$) than our active device ($2.6 \text{ L}_{\text{H}_2\text{O}} \text{ kg}_{\text{MOF-801}}^{-1} \text{ d}^{-1}$) under the same climate conditions. For example, in cycle 3, the RH difference during the adsorption phase of the adaptive device was 39% compared with 28% calculated for the active device. This is a 36% increase in water adsorption by MOF-801 during this WHC. Similarly, in cycle 3, the RH difference during the desorption phase of the adaptive and active devices were 37 and 26%, respectively, which represents a 45% increase in desorbed water by the adaptive device. Resulting from the increased WHC rates, the adaptive device realized a 44% reduction in the consumed power per WHC under low RH conditions (20%) in comparison to the active device (See Supplementary Information, Section S5). When operating at high RH conditions (46%), a 26% reduction in consumed power per WHC was achieved by the adaptive device with respect to the active device. Overall, the adaptive device consumed $1.67 - 5.25 \text{ kWh L}_{\text{H}_2\text{O}}^{-1}$ based on the climate conditions of that specific day (19 – 46% RH) and saved considerable time by requiring 1.5 cycles less than the active device. Fig. 4B shows the variation of the cycling time as the RH varies during the day.

Compared with other adsorbent-driven water harvesting devices, our MOF-801-based adaptive device reduces the amount of adsorbent material required for producing enough water required to fulfill daily personal consumption needs (3.5 L) by 75, 73, and 57% with respect to devices based on super moisture-absorbent gels, super hygroscopic polymer films, and hygroscopic salts in a hydrogel-derived matrix, respectively (Table 1).²⁶⁻²⁸ Furthermore, our adaptive device drastically reduces energy consumption by 60% compared to the benchmark device based on super hygroscopic polymer films while maintaining a smaller physical footprint (Table 1).”

Reviewer #2

1. *This manuscript reported an environment-adaptive atmospheric water harvesting (AWH) strategy by tuning the working time of the adsorption and desorption processes according to the external RH to improve the water yield per day. The whole system is composed of an air intake compartment, a sorption compartment, and a condensation compartment. Benefiting from the optimized cycling time, the designed system can efficiently generate clean water from arid environments. While the proposed system design is somehow interesting in terms of operation optimization based on an environmental adaptive strategy, the novelty and significance of this work are rather low, since neither new scientific insights of the AWH process nor an in-depth understanding of the underlying mechanism (such as thermodynamics) was proposed. In addition, the lack of detailed explanation or discussion on some critical questions and the insufficient performance comparison with literature lead to unconvincing claims of the record high water production. Overall, this work presented an engineering optimization rather than a scientific innovation. Therefore, this current version of this work is not adequate for publishing in a prestigious journal like Nature Communications, maybe more suitable for a specific engineering journal.*

Answer: We appreciate the reviewer's criticism of our work as it helped to improve the clarity of the discussion and presentation of results in the manuscript. We have performed numerous additional experiments and, based on the results of these, we have endeavored to revise the manuscript and supporting information, accordingly. We respectfully hope that these revisions address and satisfy the reviewer's concerns.

2. *The authors stated that "... an adsorption phase is defined as simply pulling air by the fan from the external environment through all three compartments ... a desorption phase is defined similarly to the adsorption phase but with the electric heater turned on ...", which means even during the desorption, continuous external air will be injected into the whole system. Since the continuously injected external air also reaches the condensation part and being condensed, how this part of inlet air contributes to the water yield? A control experiment identical to the desorption experiment but without MOF needs to be conducted to reveal the actual contribution of the MOF-801 to the water yield.*

Answer: Water vapor in the air can be condensed without any adsorbent material present. This is the basis of water harvesting via direct condensation. The challenge with this mode of operation, especially in hot, low-humidity desert climates, is the energy requirements needed to generate the requisite dew point temperature. Power consumption can be reduced by employing a MOF whose water sorption properties suit the chosen environment. An experiment was conducted in environmental conditions of 22 °C and 30% RH to conclude a 55% reduction in energy consumption when employing MOF-801. This is because the MOF concentrates the water vapor, which, in turn, raises the dew point temperature.

We have conducted an additional experiment that demonstrates the difference in the power consumption of our reported device with and without MOF-801. Specifically, at 60% RH with the condenser set at 8-11 °C (i.e., dew point for when MOF-801 is used to harvest water at 15-30% RH), the device without MOF-801 produced less than 2 mL of water. We then increased the quantity of MOF used in the device from 0 g to 100, 200, and 400 g and witnessed significant increases in water production as a function of environmental RH (20, 30, and 40% RH).

We revised the manuscript to include a discussion on the results of this comparison (Page 7, Right Column, Paragraph 3):

"Specifically, the device was gradually loaded with MOF-801 starting from 0 g (empty device) to 100, 200, and 400 g and subsequently operated under controlled climate conditions of 20, 30, and 40% RH. Water production for the device loaded (or not) with each amount was measured 3 times for 24 h for each controlled climate condition with the average output presented in Supplementary Information, Fig. S32. At 50% RH and 400 g_{MOF-801}, the device produced 870 mL_{H₂O} compared with 420 and 190 mL_{H₂O} when using 200 and 100 g_{MOF-801}, respectively. When the device was empty (0 g of MOF-801), the condenser was set to 8 – 11 °C (i.e., the dew point when using MOF; Fig. 2), which led to a water production of <2 mL. A similar trend is observed for the other measured conditions. Therefore, the dependence of MOF quantity on water production was clearly observed for the other measured conditions and the dependence of MOF quantity on water production was established."

The data from this experiment can be found in the revised Supplementary Information, Section S5, Figure S32.

Finally, the data comparing the energy with and without MOF can be found in the revised Supplementary Information, Section S3, Tables S4 and S5.

3. *While most works in literature use a closed system during desorption (easy to reach a high RH for collection), the current work chooses to maintain an open system during desorption. Please compare the difference between the two modes and provide corresponding data if necessary.*

Answer: The reviewer raises an important point. The condensation compartment during desorption is semi-open to the atmosphere via a 5 mm hole that is used for pressure release and to prevent back pressure, friction losses, and pressure drop. Indeed, this hole ensures a continuous flow of the incoming hot air and forces the condensed water to the mineralization and filtration unit. When this hole was closed, back pressure resulted, which forced the air back through the air intake compartment. Furthermore, we observed that the adsorption and desorption times increased significantly by 12-27 and 14-32 min (depending on external RH), respectively, and the water production was reduced by 7-19 mL cycle⁻¹.

Existing passive devices do not apply pressure because of the long time available at night and day to carry out the adsorption and desorption, respectively. This means condensing water vapor can happen in a closed system (see *Sci. Adv.* **4**, eaat3198 (2018)). In active devices, fast sorption kinetics are required meaning that the device must be either semi-open (for reasons described above) or be able to create a vacuum to ensure proper desorption from the adsorbent material (see *ACS Cent. Sci.* **5**, 1699-1706 (2019)). There are considerable energy input considerations for the latter, which is the reason for our semi-open design.

To clarify this point, we revised the manuscript accordingly (Page 3, Right Column, Paragraph 3):

“The condenser is fit within a funnel that is used to collect the condensed liquid water and transport it to the mineralization and filtration unit below (Fig. 1). A 5 mm outlet hole was incorporated at the bottom of this compartment to prevent the formation of back pressure, friction loss, and pressure drop. Additionally, this hole not only ensures a continuous flow of incoming hot air, but also forces the condensed water to the mineralization and filtration unit.”

A further revision was made in the manuscript (Page 5, Left Column, Line 43):

“We note that closing the 5 mm outlet in the condensation compartment forces air back to the air intake compartment, leading to a 12 – 27 min increase in the adsorption time.”

A final revision was made in the manuscript (Page 6, Left Column, Line 37):

“Similar to the adsorption phase, when closing the 5 mm outlet in the condensation compartment, a 14 – 32 min increase in the desorption time was observed, which resulted in a reduction of 7 – 19 mL cycle⁻¹ water production.”

4. *In Fig 3, the dew point at the condenser compartment was used to monitor the sorption and desorption processes along with time. At the equilibrium state (no matter during the adsorption or desorption phase), no vapor will be captured or released by the MOF-801 (no net vapor flux), which means the air reached in the condenser compartment should be the same as the external air. However, unlike the desorption phase where the inside dew point is finally equivalent to the external dew point, the inside dew point at equilibrium during the adsorption phase shows a clear difference with the inlet air. Please explain why.*

Answer: Figure 3 is a snapshot of the full adsorption behavior with an extrapolated equilibrium state to show the long adsorption time. We have revised Fig. 3A to remove the extrapolated part (Page 5, Right Column, Figure 3).

For the utmost clarity, we have added the full adsorption time behavior in a real-time experiment to the revised Supplementary Information, Section S4, Figures S27 and S28.

5. *Please explain why the authors choose different conditions between adsorption and desorption in Fig 3? The different conditions make it hard to compare the dew point change during these two connected processes.*

Answer: To develop the ‘adaptive water harvesting’ algorithm, real-life extreme environmental conditions (i.e., stress tests) were required. For adsorption, this meant high temperature (35-40 °C) and low RH (15-26%). For desorption, low temperature (20-25 °C) and high RH (14-45%) were studied. If a comparison of the performance change during adsorption and desorption phases carried under the same conditions (i.e., 1 water harvesting day) is of interest, please refer to Fig 5B.

6. *In Table S1, three kinds of procedures were studied in the active device experiments and the results show more adsorption/desorption cycles in a water harvesting cycle (WHC) leads to a higher water yield per day. The authors chose the two adsorption/desorption cycles mode (i.e., Procedure 3) as the final procedure to do the following experiments. Why didn't the authors further increase the adsorption/desorption cycle numbers? Will the water yield per day keep rising along with the increased adsorption/desorption cycles? Will it lead to a higher water yield even than the adaptive strategy?*

Answer: To properly address this valid point, we performed an additional experiment whereby we increased the number of adsorption and desorption cycles. Upon increasing the number of cycles from 2 to 3, water production increased slightly from 1.25 to 1.58 L at 10-30% RH and 2.33 to 2.97 L at 40-60% RH, respectively.

The increased water production as a function of increased cycles does not come without significant disadvantages. To carry out more cycles means that more energy will be required.

Indeed, we found that the energy consumption increased significantly from 3.51 kWhr L⁻¹ to 4.73 kWhr L⁻¹ at 10-30% RH and 7.87 kWhr L⁻¹ to 9.24 kWhr L⁻¹ at 40-60% RH.

It is safe to conclude that although increasing the number of cycles may increase water output, the corresponding increase in energy consumption outweighs those benefits.

The data from this new experiment can be found in the revised Supplementary Information, Section S3, Table S1.

7. *In Fig 4, why the curve of active operation didn't start from 0 min? The current figure is hard to read, and a summary figure compares the time per cycle and cycling numbers per day between the adaptive and active operations will help the reader to grasp their difference. In addition, an energy efficiency analysis may also be helpful in showing the advantages of the adaptive strategy over the normal active strategy.*

Answer: Figure 4 expresses the RH response behavior of each device from mid-day into the night. The timing is arbitrary. The experiment was conducted using two identical devices each loaded with 400 g of MOF-801 and operated under the same controlled environment. At the beginning of the figure (mid-day), both devices processed air having low RH. A clear time difference (46.5 min) was observed between the adaptive and active device, which is expected as the adaptive device requires less time per cycle when compared to the active device at low RH. As RH increases throughout the day into the night, the adaptive device starts to require more time and produce more water leading to a shortened time difference (25 min at cycle 5). When the RH is at its highest during the night, the time difference becomes indistinguishable (<3 min at cycle 9) between the adaptive and active devices.

We revised the manuscript to clarify this point (Page 6, Right Column, Paragraph 2):

“To evaluate the performance of this adaptive mode of operation against the previous active mode, we loaded two identical devices with 400 g of MOF-801 and programmed the operation of each device according to Figs. S26 and S31 in the Supplementary Information. Under the same climate conditions, the performance for both devices in terms of change in RH as a function of time is presented in Fig. 4A for a 14 h total time spanning a portion of the day into the night. Over this period, the lowest and highest external RHs were measured to be 19 and 46%, respectively. Interestingly, the first cycle of the adaptive device with external climate conditions of 23% RH and 25 °C was found to be 46.5 min faster than the active device demonstrating its ability to adapt and enhance the WHC rate.

When considering an adsorbent material's water uptake capacity is limited by RH regardless of extended cycling time, the advantages of the adaptive mode of operation become clear. Under low RH conditions, the adaptive device will reach its maximum capacity at the specified RH, at which point it will immediately begin the desorption phase without consuming more time in the adsorption phase. However, for the active device in these same conditions, the adsorbent will reach its maximum capacity faster. Therefore, the remaining pre-programmed time for the adsorption phase is wasted. Similar reasoning can be extended to the desorption phase. Lower

water uptake per cycle in the adaptive mode of operation requires a shorter desorption time than the pre-programmed desorption time used in the active mode of operation (Fig. 4C).

As the RH increases throughout the day and into the night, the difference in cycling rates becomes indistinguishable (e.g., <3 min for cycle 9 at 46% RH and 25 °C). This is because an increased RH during the night increases MOF-801's water uptake, which, in turn, increases the timings of the adsorption and desorption phases of the adaptive device (Fig. 4A). Under high RH conditions, the adaptive mode of operation will reach MOF-801's maximum capacity and, once achieved, will immediately start the desorption phase. For the same conditions, the pre-programmed timing for adsorption phase of the active mode of operation will not be enough to reach MOF-801's maximum capacity leading to lower water production (Fig. 4D)."

8. *The figure caption of Fig 4 is inconsistent with the figure. Please correct it and also check the whole manuscript to avoid any potential errors.*

Answer: We have revised the caption of Figure 4 in the revised manuscript.

9. *The author claimed "a 169% increase in water productivity and a 44% reduction in power consumption compared to state-of-the-art reported devices" in the conclusion section. Therefore, a comprehensive and systematic comparison with literature is necessary to support this claim.*

Answer: We have separated the water production outputs from low (10-30%) and high (40-60%) RH atmospheric conditions. Our adaptive device produces 1.8 L Kg⁻¹ day⁻¹ at low RH compared with 0.7 L Kg⁻¹ day⁻¹ for the best-performing active device. This represents a 157% increase in water production at low RH. Furthermore, our adaptive device produces 3.5 L Kg⁻¹ day⁻¹ at high RH compared with 1.3 L Kg⁻¹ day⁻¹ for the benchmark active device. Indeed, this represents a 169% increase in water production at high RH.

For the energy requirements, the benchmark active device produces 0.7-1.1 L Kg⁻¹ day⁻¹. To produce this amount, 25 cycles were carried out over 72 h, which equates to 8.33 cycles per day. Each cycle consumes 90 W and the length of each cycle is 2.88 h. This means that each cycle consumes 259 Whr. Multiplying this value by the 8.33 cycles per day equals 2,158.33 Whr daily. Based on an average of 0.9 L Kg⁻¹ day⁻¹, producing 1 L of water consumes 2398.14 Whr. Compared with the 1670.27 Whr required for our adaptive device, our report represents a 44% reduction in consumed power.

Reviewer #3

1. *The powder size and packing efficiency of MOF-801 should be presented since they are related to the timing of adsorption and desorption.*

Answer: To determine the crystalline powder's exact size, we have carried out scanning electron microscope imaging.

This new data can be found in the revised Supplementary Information, Section S2, Figure S24.

We agree that for maximum mass transport efficiency, the MOF powder should be packed in the sorption compartment in the form of shaped bodies. However, over the course of this research program, our attempts to shape MOF-801 using different binders and manufacturing processes led to significant drops in porosity and water sorption properties. We, therefore, decided to simply use MOF-801 powder directly on the sorption compartment trays.

We revised the manuscript to point this out in the concluding section as a much-needed future direction (Page 10, Right Column, Paragraph 2):

“Moving forward, maximizing heat and mass transfer via optimized airflow dynamics (e.g., fluidized bed, shaped bodies) will lead to further exploitation of the sorption properties of MOFs for practical water harvesting.”

2. *Is the power of the fan influencing the adsorption and desorption efficiency?*

Answer: Yes, the more powerful the fan used, the greater the amount of air that can be processed, which, in turn, enhances the adsorption and desorption efficiency. Accordingly, we used the most powerful fan possible that would not lead to dispersal of the MOF-801 powder from the sorption compartment trays.

We revised the manuscript to point this out in the concluding section as a much-needed future direction (Page 10, Right Column, Paragraph 2):

“Moving forward, maximizing heat and mass transfer via optimized airflow dynamics (e.g., fluidized bed, shaped bodies) will lead to further exploitation of the sorption properties of MOFs for practical water harvesting.”

3. *I am confused about the condensation compartment dew point decrease during the adsorption process if the fan is open and the system is connected to the outer space, and the system is still open during the desorption process?*

Answer: The answer for this is provided in Comment #3 for Reviewer #2.

4. *How does the environment temperature influence the harvesting process?*

Answer: Temperature and RH are the two most important properties for atmospheric water harvesting. These properties combine into the dew point, which reflects actual water content in the air. In a water harvesting device setting, we can consider the following to be true:

- Assuming constant RH, increased temperature = more water in the air
- Assuming constant RH, decreased temperature = less water in the air
- Assuming variable RH, increased temperature = less water in the air
- Assuming variable RH, decreased temperature = more water in the air

Because of points 1-4, adaptive water harvesting is necessary!

5. *The caption of figure 4 is not consistent with the figure.*

Answer: We have revised the caption of Figure 4 in the revised manuscript.

REVIEWERS' COMMENTS

Reviewer #2 (Remarks to the Author):

The authors have addressed all the questions I raised properly. I would support the acceptance of this work now.